# Evaluation of a training program for rheumatic heart disease screening integrated into the public health system in Uganda

Paul W. Warren[1,2]*, Scott Wirth[3], Jafesi Pulle[4,5], Doreen Nakaagayi[4,5],
Charles Oyoo Akia[6], Evelyn Ayaa[7], Michaela Pardo[8], Francis Odong[4], Ronald Ogwanh[4],
John Omagino[4], Isaac Otim[4], Craig Sable[9], Alison Spaziani[9], Henry Okello Otto[7],
David Watkins[8], Emmy Okello[4], Kristen Danforth[8], Andrea Beaton[1,2]

**1** The Heart Institute, Cincinnati Children's Hospital Medical Center, Cincinnati, Ohio, United States of America, **2** University of Cincinnati College of Medicine, Cincinnati, Ohio, United States of America, **3** Division of Pediatric Cardiology, University of Utah School of Medicine, Salt Lake City, Utah, United States of America, **4** Uganda Heart Institute, Kampala, Uganda, **5** Makere University, Kampala, Uganda, **6** Ministry of Health, Kampala, Uganda, **7** Kitgum District Health Department, Kampala, Uganda, **8** University of Washington, Seattle, Washington, United States of America, **9** Children's National Medical Center, Cardiology, Washington, District of Columbia, United States of America,

* paul.warren@cchmc.org

## Abstract

### Introduction

Echocardiography screening for rheumatic heart disease (RHD) has gained support as a public health approach, but scale up of RHD screening services is complex. We sought to evaluate the effectiveness of a novel training program to build non-expert competency for RHD echocardiography screening within the Uganda public health system and to describe the human and material resources required to support it.

### Methods

Guided by a logic model, we evaluated the Accelerating Delivery of Rheumatic Heart Disease Prevention in Northern Uganda (ADUNU) Program, a novel RHD control program, 15 months after its implementation within the Ugandan public health care system.

### Results

Sixty-one healthcare workers (HCW) across 10 public health facilities started in training under the program, of which 58 (95%) advanced past the initial stage of training and earned conditional certification to screen for RHD with ongoing remote and in-person feedback and oversight. Of these, 17 (29%) completed all stages of training and earned full certification to independently screen for RHD with no ongoing oversight. A total of 17,927 community members were screened through ADUNU during the program's first 15 months. After receiving final certification, 14 HCWs (93%)

**Data availability statement:** All relevant data are within the paper and its Supporting Information files.

**Funding:** NIH Grant R01HL164615.

**Competing interests:** The authors have declared that no competing interests exist.

**Abbreviations:** ACEP: American College of Emergency Physicians, AI: Artificial intelligence, ADUNU: Accelerating Delivery of Rheumatic Heart Disease Prevention in Northern Uganda, CI:Confidence Intervals, DHO: District Health Office, HCIII: Health Center III, HCW: Healthcare worker,PLAX: parasternal long axis,RHD: Rheumatic heart disease,RRCU: RHD Research Collaborative in Uganda,UHI: Ugandan Heart Institute,WHF– World Heart Federation.

continued to perform screening echocardiograms (≥20/month) at median follow-up of 8 months [IQR 8–10]. HCW sensitivity and specificity were 61% and 96%, respectively.

## Conclusion

Development and deployment of a large scale RHD screening echocardiography training program within an existing public health system is feasible. Future program iterations are needed to improve HCW screening sensitivity and decrease the reliance on human resources.

---

## Introduction

Echocardiography is a powerful tool for early rheumatic heart disease (RHD) detection [1–3]. While originally employed as epidemiological research, echocardiography screening for RHD has more recently gained support as a public health approach [4–6]. This shift resulted from a randomized controlled trial which found early diagnosis and initiation of secondary antibiotic prophylaxis significantly decreased the 2-year risk for RHD progression [7]. In 2023, the World Heart Federation (WHF) published updated guidance on echocardiographic screening for RHD in high-risk populations, and in 2024 the World Health Organization formally recommended countries implement screening, especially in high prevalence settings [1,8].

However, scale up of RHD screening services is complex, especially in resource-limited settings with low availability of subspecialty expertise and equipment [9]. Shifting the screening burden to a more readily available workforce through task-shifting presents an alternative solution. Training for task-shifting RHD screening has been proposed since at least 2015, but no standardized programs have emerged [10]. Most models have relied on training non-expert health care workers (often nurses), utilizing hand-held ultrasound machines [11–14], and performing simplified screening protocols to reduce time and complexity [15–17]. These approaches have shown acceptable sensitivity and specificity. However, most studies have involved a small number of non-expert providers, often in the context of an incentivized research study. With RHD screening now recommended as a public health approach, there is need for replicable training programs to support the scale-up of service provision [18].

In 2023, the Ugandan Ministry of Health and the Uganda Heart Institute (UHI) embarked on the deployment of an evidence based RHD control program, supported by the RHD Research Collaborative in Uganda (RRCU) as an implementing partner [19]. The Accelerating Delivery of Rheumatic Heart Disease Prevention in Northern Uganda (ADUNU) Program aims to find more cases of RHD by integrating RHD screening into the existing Ugandan public health system by offering screening at all community-level health center III (HCIII) facilities in participating districts [19].

At the end of the first year, we evaluated the effectiveness of the ADUNU training curriculum at building non-expert HCW competency in RHD echocardiography screening and HCW integration of screening into their clinical practice. To do so, we

utilized a logic model to map the program's resources, actions, outputs and outcomes [20,21], with a goal of understanding how this program should be modified to create a standardized training program that could be adopted globally.

## Materials and methods

### Evaluation timing

Fifteen months after the initiation of the ADUNU training curriculum, we utilized programmatic data to 1) describe the human resources, material resources, and activities that were input into the training program, 2) describe the outputs of the training program, and 3) evaluate the outcomes of the training program. Data sources included training team field notes, meeting notes, training logs and certifications, echocardiographic screening logs, and reports of image agreement between HCWs and experts who were providing remote support. Program data on the training program for RHD screening collected between May 2023 and August 2024 was utilized for the evaluation. Data was accessed for research purposes from 22/9/24–3/3/25.

### Ethical approval

Programmatic data utilization for ADUNU is approved under the Uganda Heart Institute Institutional Review Board (UHIREC0008/UHIREC0021), the Ugandan National Council of Science and Technology (HS2470ES), and the Cincinnati Children's Hospital Medical Center Institutional Review Board. No individually identifiable data was utilized for this evaluation and no individual consent was required. The funder of the study had no role in study design, data collection, data analysis, data interpretation, or writing of the report.

### Setting

Kitgum District, a largely rural district in northern Uganda, was selected by the Ugandan Ministry of Health for a pilot of the ADUNU program (Fig 1). Kitgum is located approximately 460 km north of Kampala, Uganda's capital city. The population of Kitgum was estimated to be 226,700 in 2020 [22]. There are 11 government sub-country or higher level health centers in Kitgum district, which include a District Hospital, one health center IV, and nine health center IIIs. HCWs, Nurses and clinical officers, in these facilities who were interested in participating in the program and available on the day of training were eligible for RHD screening training under the ADUNU program. Onboarding of participating health care centers included district engagement that started nine months prior to HCW training. Training preparation included identification of facilities for training by the District Health Office (DHO), messaging about the training from the DHO office to the health facilities, and intake at each of these facilities to identify staff that qualified for participation.

### Training plan

The training plan for RHD screening had three components including initial training (didactic and hands-on), audit and feedback, and in-person booster sessions as needed for re-training. Two benchmarks, conditional and final certification, were used to assess whether HCWs reached a satisfactory level of screening competence.

For initial training, tablets pre-loaded with a set of ten training modules were delivered to participating clinics at least two weeks prior to in-person training. Each trainee was asked, but not required, to complete these modules to build background knowledge. This was followed by a 1-week in-person training, delivered by the UHI technical team. In-person sessions included interactive lectures and proctored hands-on practice screening. Trainees were taught to use the Lumify device (Philips Healthcare, Best, Netherlands) and focused on mastery of a three-step parasternal long axis (PLAX) protocol consisting of a two-dimensional interrogation of the mitral and aortic valves, color Doppler interrogation of the mitral valve, and color Doppler interrogation of the aortic valve. Interpretation criteria for a positive echocardiogram were practiced, including mitral regurgitation ≥ 2 cm, aortic regurgitation jet ≥ 1 cm, restricted mitral valve opening (identified

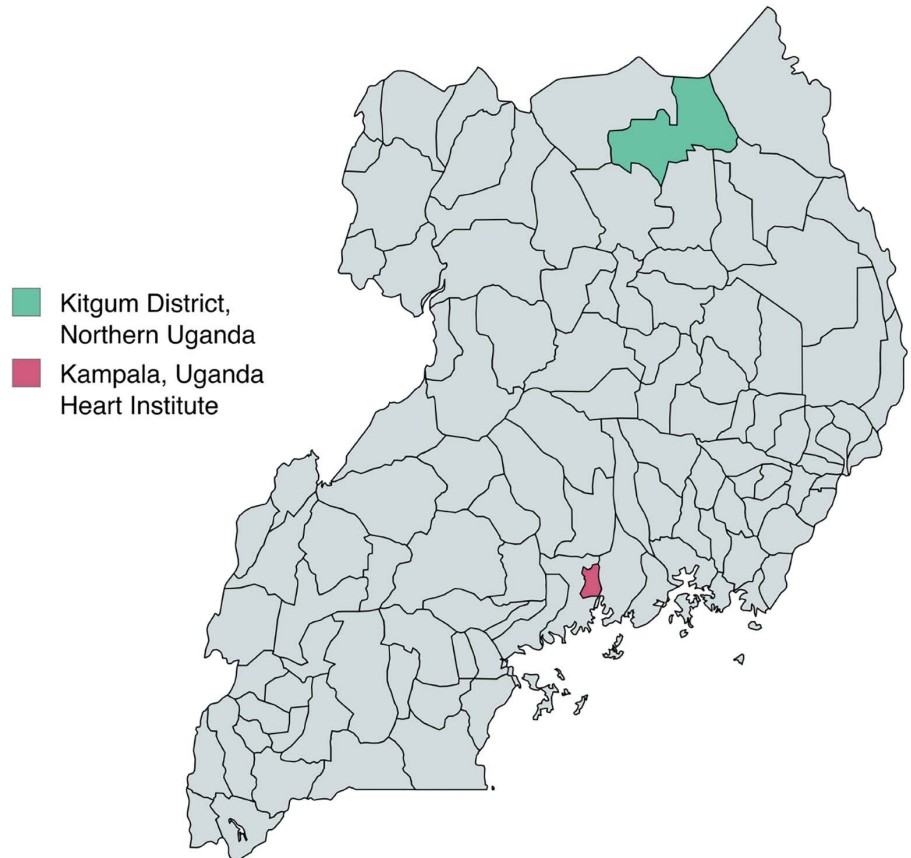

**Fig 1. Map of Uganda Showing ADUNU District Kitgum in Relation to the Uganda Heart Institute in Kampala.** Reprinted under a CC BY license, with permission from the Kitgum District Health Officer, original copyright 2025.

by turbulent color Doppler inflow), pericardial effusion ≥ 1 cm, or qualitatively reduced systolic left ventricular function. To ensure clinical safety, positive studies (either from HCW or expert) were referred for confirmatory echocardiogram at a regional health center. The full outline of the training program can be found in S1 Fig.

After initial training, audit and feedback was provided through remote support. In brief, clinics uploaded all screening studies to a cloud-based repository (Tricefy, Trice Imaging, Del Mar, California, USA). A team of six cardiologists in the US and Uganda reviewed the echocardiograms, utilizing the same criteria to determine if studies were positive or negative [12]. Image quality was assessed by the American College of Emergency Physicians (ACEP) ultrasound image quality grading scale [23]. ACEP scores included: (1) no recognizable structures, no objective data can be gathered, (3) minimum criteria met for diagnosis, recognizable structures but with some technical or other flaws, or (5) minimal criteria met for diagnosis, all structures imaged with excellent image quality and diagnosis completely supported. Reviews were performed every two weeks with the results communicated back to trainees by the ADUNU district team to provide continuing education, including intermittent group review of missed cases.

Booster sessions were provided for trainees who continued to perform screening echocardiograms but had not yet completed training by 3 months. These in-person sessions were customized to the learner's needs but included on-site (at the trainee's home clinic) and off-site (RHD training center at the District Hospital) sessions to address gaps in knowledge and performance, with a heavy focus on bolstering recognition of positive studies.

## Trainee evaluation

There were multiple opportunities for trainee evaluation. At the end of the 1-week in-person training, HCWs who attended at least 80% of training sessions and completed at least 10 independent scans during the training were assessed for conditional certification. Successful conditional certification included trainees completing 2 scans with direct observation of basic machine skills (turn on, set up for a study, hold probe correctly), correct patient positioning on the bed, and adequate image acquisition (could independently acquire three step PLAX protocol). Achieving conditional certification allowed trainees to perform independent RHD screening at their health facility, with audit and feedback oversight. If conditional certification was not achieved at the end of the first week, a customized plan was put into place to include additional mentorship from the ADUNU technical team and/or peer to peer mentoring at the facility. Repeated evaluations were available at later dates when the ADUNU technical team was conducting regular clinic site visits. Final certification was assessed on a rolling basis. Minimum criteria for final certification included completion of at least 100 independent scans, scanning for a minimum of 12 weeks, and correct identification of >90% screen positive scans for 2 consecutive weeks. Achieving final certification allowed health care workers to perform independent RHD screening at their facility, without the need for direct audit and feedback, although ongoing support remained available if needed.

## Logic model iteration (Fig 2)

We used a logic model framework, a process tool for program planning, implementation, and evaluation, to illustrate the local program and its various components including inputs, activities, outputs, and outcomes [24,25]. The first author (P.W.) generated a logic model 15 months after program implementation using data of programmatic documents (described above) and assessed whether these components were aligned with the intended outcomes. The logic model was reviewed and revised through an iterative process with the remainder of the research team.

## Outcomes and statistical analysis

The overall goal of this analysis was to evaluate if the ADUNU training program resulted in trained HCWs who were successful at integrating RHD screening into practice. We tabulated descriptive statistics to quantify resources, activities, and outputs of the ADUNU training program. Three primary outcomes were identified. First, we were interested in the

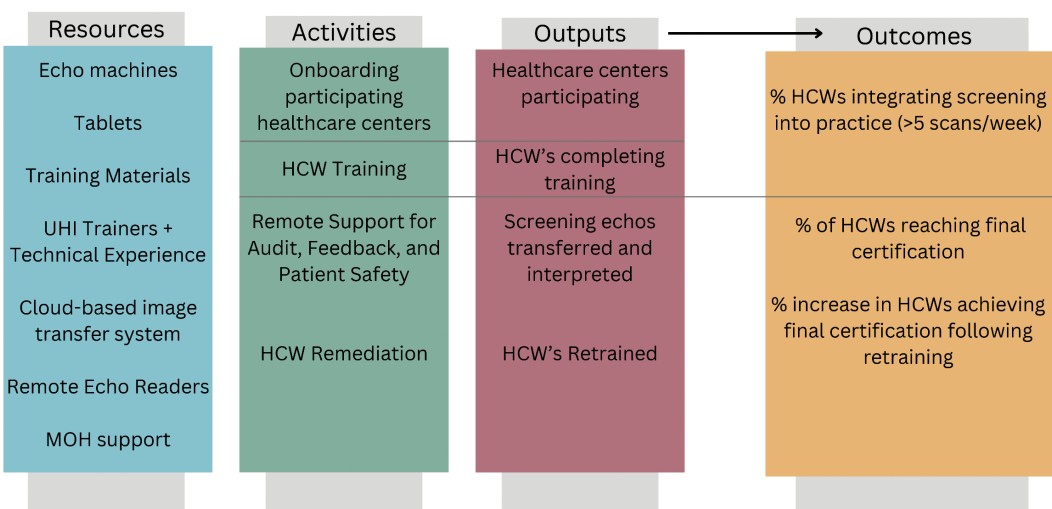

**Fig 2. A logic model for planning and evaluation of the ADUNU Training Program for RHD Screening.**

percentage of HCWs who integrated screening into practice, defined as completion of an average of at least 20 screening studies per month. This was further broken down by participating facility. Second, we measured the percentage of HCWs who reached final certification, including the time it took for them to reach final certification and the number of screening studies completed prior to final certification. Finally, we looked at a sub-analysis of the percentage of HCWs that required a re-training (booster session) to achieve final certification.

We tabulated descriptive statistics for screening and agreement data. Statistical analysis was performed using RStudio (R version 4.5.2) [26]. We report the key findings from the local program document review in the logic model framework as inputs, activities, outputs, and outcomes.

## Results and discussion

### Resources

Two main categories of input were necessary for training: supplies/equipment and human resources, which includes quantified time as well as program knowledge, experience, and expertise. Supplies were either purchased (tablets, generators, and training workbooks) or provided in-kind (echo machines, training modules, and cloud-based image storage/transfer). Table 1 contains a comprehensive list with estimated values for in-kind inputs. Human resources for training included a two-person team of trainers from the RRCU and the district RHD focal person, a health care worker previously trained to provide comprehensive RHD care for the district. Training was supported in some phases by a remote core echocardiography lab and six remote expert echocardiography reviewers.

### Activities

During the initial training week, 61 trainees spent an average of 26 hours each participating in didactic and hands-on scanning sessions (Table 2). The RRCU trainers and the RHD district focal person spent a combined 135 hours during each initial training week at each of the 11 sites, followed by an additional 40 hours/week providing ongoing audit and feedback. HCW remediation took place for those who were meeting their screening volume targets but not achieving final certification (seven to eleven months after initial training). There were 29 trainees who participated in these one-day retraining sessions, which included PowerPoint review of positive and missed positive echo clips and hands on image acquisition and interpretation. The cumulative cost of the initial training, audit and feedback, and remediation activities was $65,574.

### Outputs derived from program resources and activities

In total, all 10 qualifying public facilities participated in training, including one health center IV and nine health center IIIs. Initial training was completed by 61 HCWs. Most HCWs (98%) attended at least 90% of the in-person didactic lectures.

Table 1. Supply and Equipment Resources Investment for RHD Screening Training under the ADUNU Program.

|  | Description | Units | ADUNU Cost | Actual Cost |
|---|---|---|---|---|
| Echo Machines | Lumify Probe (Philips Healthcare), $4500/probe | 15 Probes | In-kind | $67,000 |
| Tablets | Samsung Galaxy S9, $349/tablet | 16 | $5584 | $5584 |
| Generator(s) | Required during training to provide charging for devices on-site, $250/unit | 2 | $500 | $500 |
| Training Materials: Modules + Presentations | Moodle (open source) software (free), PowerPoint (Microsoft) included on trainer computers (free) | n/a | $0 | $0 |
| Training Materials: Hard-copy workbooks | Printing of hard-copy custom workbooks ($6.20/workbook) | 65 | $405 | $405 |
| Cloud-based Image Transfer | Tricefy (Trice) cloud-based image management and documentation solution | 1 | In-kind | $12,581 |

Table 2. Human Resource Investment for RHD Screening Training under the ADUNU Program.

| | Description of Human Resources | | | |
|---|---|---|---|---|
| RRCU Trainers | 3 staff, 1 adult cardiologist, 2 senior research nurses ((≈19.20/hour*) | | | |
| RHD Focal Person | Fulltime MOH senior nurse RHD district coordinator (≈5.57/hour*) | | | |
| Trainees | Public health nurses, midwives, and clinical officers (≈2.92/hour*) | | | |
| Remote Echo Readers | 6 cardiologists/cardiology fellows, US pediatric cardiology median salary (≈$98.50/hour*) | | | |
| Core Echo Lab Coordinator | Research coordinator that cataloged studies coming in for reading, assigned studies to be read, and compiled and sent reports for local team action (≈$28.85/hour*) | | | |
| HCW Training | 1 week/health center, 6 hours of didactics dedicated to RHD screening, ≈ X hours of hands-on supervised scanning practice/trainee | | | |
| RRCU Trainers | 2 per training week, at health center ≈9 hours/day for 5 days, at 11 health centers | 990 hours | $19,008 | $19,008 |
| RHD Focal Person | During training week, at health center ≈9 hours per day x 5 days | 495 hours | $2757 | $2757 |
| Trainees | 61 trainees, each with 6 hours of didactics, ≈ 20 hours/scanning or observing scanning/week | 1586 hours | In-kind | $4631 |
| Audit + Feedback | Health center upload to cloud, core echo lab coordinator, remote echo reader overread, went on for X weeks. | | | |
| RRCU Trainers | 1 in-district team member providing feedback to sites with in-person visits and WhatsApp reporting, approximately 20 hours/week for 52 weeks. | 1040 hours | $19,968 | $19,968 |
| RHD Focal Person | 1 in-district team member providing feedback to sites with in-person visits and WhatsApp reporting, approximately 20 hours/week for 52 weeks. | 1040 hours | $19,968 | $19,968 |
| Remote Echo Readers | Approximately 3 minutes per review and reporting x 6609 studies | 202 hours | In-Kind | $19,927 |
| HCW Remediation | 6 retraining events, each 1 day in length, combination of on-site (at a healthcare center) and offsite at the District Hospital | | | |
| RRCU Trainers | 2 per training day, at health center ≈9 hours/day for 6 days | 108 hours | $2,074 | $2,074 |
| RHD Focal Person | 1 per training day, at health center ≈9 hours/day for 6 days | 54 hours | $1,037 | $1,037 |
| Trainees | 29 trainees, at health center ≈9 hours/day for 1 day | 261 hours | $762 | $762 |

*Based on a 40-hour work week.

Abbreviations: HCW – Health Care Worker, RRCU – Rheumatic Heart Disease Research Collaboration in Uganda

Conditional certification was granted to 58 (95%) HCWs at the completion of their initial training, indicating sufficient baseline image acquisition skills (Table 3).

**Audit and feedback and retraining.** Of the 10,232 echos performed during the audit and feedback period, 9,482 (93%) were successfully uploaded to the cloud for remote interpretation. The most common technical challenges included 477 (5%) studies that were not uploaded due to WIFI disruption and 273 (3%) studies that were deleted or not saved prior to image upload. The major areas of feedback included optimization of image acquisition (placement of color box over the correct valve, appropriate image depth, appropriate image 2D and color gain), troubleshooting of image upload/technical problems, and recognition of RHD positive scans.

Twenty-nine HCWs (54%) required and completed at least one remediation session. HCWs were eligible for remediation if they had been conditionally certified for at least three months and had performed at least 50 independent scans, but had not yet met final certification criteria. There were four additional HCWs who met criteria for remediation but missed the remediation due to illness, inability to leave their health center, or conflict with personal time off.

### Outcomes derived from programmatic outputs

**HCWs achieving RHD echo screening competency.** Of the HCWs who reached conditional certification, 29% (17/58) (29%) achieved mastery of the required set of competencies and received final certification. The median duration to achieve final certification was 6.9 months (IQR 5.0–8.2). Four HCWs reached final certification without the need for

**Table 3.  Outcomes for RHD Screening Training under the ADUNU Program.**

| | Description | |
|---|---|---|
| HCWs Participating in Initial Training | Defined as those who started the initial training week | 61 (Total N) |
| HCWs Reaching Conditional Certification | Defined as successfully completing the initial training | 58 (95.1%; 95% CI 86–98.9%) |
| | | 58 (Total N) |
| HCWs Integrating Echo Screening into Practice | Defined as ≥20/month from completion of initial training to final certification or study completion | 23 (39.7%: 95% CI 28.1–52.5%) |
| HCWs Reaching Final Certification | Criteria include ≥100 independent scans, scanning for ≥12 weeks, and correct identification of ≥90% screen positive scans for 2 consecutive weeks. | 17 (29.3%; 95% CI 19.1–42.1%) |
| HCWs Continuing Screening After Final Certification | Defined as ≥20/month from receipt of final certification to study completion | 14 (93.3%; 95% CI 68.1–99.8%)* |
| Screening Echos Completed After Final Certification | Echos performed after HCW achieved final certification | 7,294 |

*2 HCWs reached final certification at the time of data analysis and were therefore not able to be assessed for continued screening, making this 14/15 HCWs. The lone HCW not screening ≥20/month from receipt of final certification to study completion was performing 19 scans/month.

CI = confidence interval

re-training, while 13 HCWs reached final certification after re-training. 67% (39/58) of HCWs were asked to stop screening after failing to meet a minimum number of practice scans over several months and the remaining two (3%) HCWs remained in training at the time of data collection. HCWs who achieved final certification continued to perform a median of 44 (IQR 22–58) screening echocardiograms per month through the last follow up (median 8 months IQR [8–10]).

**Sensitivity, specificity, and image quality.**  During the conditional certification period, 97% of echos with an expert overread were of adequate diagnostic quality (Table 4). HCWs achieved excellent image quality (ACEP 5) in 59% of studies, adequate image quality (ACEP 3) in 38%, and lack of diagnostic images (ACEP 1) in only 3%. Image quality improved from an average rating of 3.39 in the first 3 months to 4.06 in the last 3 months of data collection.

The overall sensitivity and specificity of nurses during the conditional certification period, based on expert overread was 61% (95% CI: 55.7–65.7%) and 96% (95% CI: 95.2–96.3%), respectively (Table 4), with those receiving final certification achieving >90% sensitivity over at least 2 consecutive weeks. HCW training was very effective at teaching HCWs how to recognize moderate and severe RHD, as evidenced by 25/27 cases of moderate/severe RHD appropriately graded screen positive during the audit and feedback period.

**HCWs integrating screening into practice.**  Of those reaching conditional certification, 23 (40%, 95% CI 28.1–52.5%)) continued to integrate screening into practice (≥20 scans/month) over the course of the study period. One likely contributor to HCWs not integrating screening into practice was a planned annual provider rotation between healthcare centers in the district, which included 27 of the 58 trained HCWs (47%). This rotation, which occurred 6–11 months into RHD screening training, resulted in some facilities having no trained HCWs, some HCWs moving to facilities that did not offer screening (i.e., lower-level facilities, HCIIs). On-boarding processes for those transferred also resulted in 4–6 weeks where trained HCWs were not screening (Fig 1). An additional barrier was a single HCIII that was temporarily closed during the training period for 4 months. This facility had 2 trainees assigned, who were not able to continue their practice Fig 3.

**Early program impacts.**  In total, 17,927 community members were screened under the ADUNU program during the 15 months included in this study. This includes 401 screening echos during the initial training, 10,232 screening echos during the audit and feedback period, and 7,294 screening echos by HCWs after achieving final certification. Of those

**Table 4. Characteristics of all RHD Screening Studies.**

| Total RHD Screening Studies | 17,927 |
|---|---|
| RHD Screen Positive | 943 (5.3%) |
| Study Quality (total graded 6,528) | |
| Excellent – ACEP 5 | 3,839 (59%) |
| Adequate – ACEP 3 | 2,495 (38%) |
| Inadequate – ACEP 1 | 194 (3%) |
| Characteristics of Individuals Receiving Screening | |
| Median age (years) | 18 years (IQR 12–27 years) |
| Female | 67% |
| Screener Sensitivity for RHD | 60.8% (95% CI: 55.7–65.7%) |
| Screener Specificity for RHD | 95.8% (95% CI: 95.2–96.3%) |
| Positive Predictive Value | 50.6% (95% CI: 46–55.3%) |
| Negative Predictive Value | 97.1% (95% CI: 96.7–97.6%) |

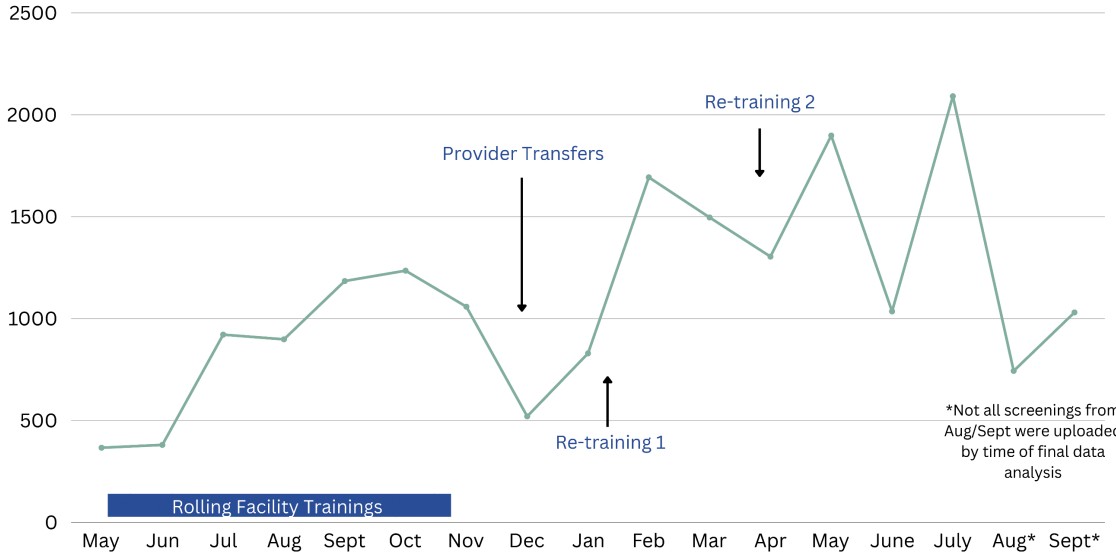

**Fig 3. Monthly screening variation in Kitgum District.**

screened, the median age was 18 years (IQR 12–27 years), with 67% being female (Table 3). Screening was positive in 5% of studies, generating 943 referrals for confirmatory diagnosis.

### ADUNU impact and key lessons learned for future program iterations

**Early success and impact.** Our evaluation of the ADUNU program's training curriculum for RHD echocardiography screening 15 months following its implementation, revealed that the RHD echocardiography training within an existing public health system in sub-Saharan Africa is feasible and effective, but resource-intensive. Ultimately, 17 HCWs achieved final certification for RHD screening with most continuing to incorporate screening into their practice at the time of this

analysis. While other large scale echocardiography screening studies have utilized task-shifting, our study is the first to critically evaluate the implementation of an RHD screening training program using a logic model lens [15,27,28]. Our study also adds to the growing regional East African literature regarding RHD screening programs [29]. Our analysis has revealed several key insights that highlight areas, which once refined, could further strengthen our training model and expand its utility to other RHD endemic global health settings.

**Need for improved RHD sensitivity.** While most HCWs demonstrated technical competency in image acquisition within the first week and maintained high image quality throughout the study, there were notable gaps in screening accuracy in particular pertaining to recognizing positive cases. Sensitivity during the conditional certification period was 61%, which falls below the 67–84% sensitivity reported in prior studies of briefly trained non-expert providers and our own programmatic target of 90% [10,15,16]. Sensitivity must be improved as delayed diagnosis and enrollment in the appropriate care-based registry may allow for illness progression and worse health outcomes. Our data shows that sensitivity can be increased through intensive re-training, specifically focusing on identifying positive cases through frequent, repetitive pattern recognition practice of RHD positive screening echocardiograms. However, this has implications for costs and feasibility, which must be taken into consideration when developing the re-training program.

**Need for decreased reliance on human resources.** ADUNU's use of a remote audit-and-feedback approach during conditional certification encountered several challenges related to delays in gathering screening images, uploading them to the cloud server, and having them overread by the remote team of cardiologists, resulting in slower than anticipated feedback to trainees. Additionally, the overread burden for this remote team was extraordinarily high. Each cardiologist overread more than a thousand screening studies, a responsibility that would be difficult, if not impossible, to replicate with larger or expanded RHD screening programs in other endemic settings. Artificial intelligence (AI) has shown promise in correctly diagnosing RHD on portable echocardiography machines and may offer an alternative, although accurate recognition on handheld ultrasound machines and single view protocols, which were used in this study, is still in development [30].

While reliance on clinicians for training and retraining is common [28], the training and re-training burden on ADUNU's was significant across the 15 months, (average time burden 40 hours per week), and even so, staff could not visit all 11 facilities frequently enough to provide consistent in-person guidance, further limiting support. Possible solutions include regular in-person booster sessions at a central training location, such as the District Hospital. However, this approach is resource-intensive and requires taking HCWs away from their daily duties. An alternative is utilizing AI to automate booster training and provide continuous education [31]. We are currently exploring innovative solutions such as leveraging large language models and just-in-time educational content delivered via WhatsApp to improve this training model, enabling more timely and consistent learning without the logistical constraints of in-person sessions.

**Identification of early adopters/high performers.** The program also revealed that only a subset of HCWs continued to screen consistently, suggesting that both facility-level and individual-level factors likely influence adoption and performance. These factors may include available resources, organizational support, personal motivation, enjoyment of screening, and the perceived value of the screening program. Ongoing research aims to identify these key determinants, which could allow for more efficient use of training resources by identifying high-performing HCWs early in the process. Furthermore, recognizing high performers could facilitate peer mentorship, fostering a collaborative learning environment that boosts overall screening adoption and performance across facilities. Ultimately, understanding these adoption determinants will help refine the training model, making it more efficient, equitable, and impactful [32].

**Training flexibility.** The rollout of the ADUNU program faced several unanticipated challenges, most notably the rotation of HCWs across facilities, which disrupted training continuity and left some facilities without adequately trained HCWs. While such a rotation system may not be common in other regions seeking to implement screening programs, it highlights the need for the innovative and automated solutions proposed above that are less reliant on in-person sessions. In the meantime, we have planned an annual training for both HCWs who did not reach final

certification in the initial round but remain interested, as well as for new HCWs who have not yet received training. This approach will help maintain a consistent pipeline of trained HCWs and minimize disruptions in service delivery. Ultimately, if this approach scales across Uganda, embedding RHD screening into standard nursing education may be a more sustainable model [33].

 **Proposed modifications for next program iteration.** The current analysis is limited by the lack of long-term data. Research is underway to assess the program's longer-term impact on district-level RHD case identification. Additionally, several modifications have been made to the training program based on our key insights. There is more frequent and repetitive HCW pattern recognition practice of positive RHD screening echocardiograms, with the understanding that increased experience and repetitions will increase sensitivity. We are leveraging individual HCW referral data (such as true and false positive scans at the confirmatory care clinic) to guide re-education instead of relying on remote audit-and-feedback to reduce the need for a cardiologist overread. And we are implementing tablet-based education that can be accessed offline at the facilities (S2 Fig) to mitigate the time burden on ADUNU staff. This model is currently being rolled out in a second district in Uganda, providing an opportunity to test these modifications aimed at improving screening uptake, sensitivity, and reducing training burdens. Prior to expanding at the regional and global level, we must create a more effective and streamlined training program. Further evaluation will be essential to understand the full impact of these changes as we continue to refine the training model.

## Conclusion

The ADUNU training model for scaling up RHD screening has proven that integrating RHD screening into the Ugandan public health primary care system is both feasible and impactful, with high uptake observed in the first 15 months of the program. While the sensitivity is currently suboptimal, the future training program iteration is designed to improve disease recognition while also reducing the burden on human resources. With the 2024 World Health Organization guidelines now recommending routine RHD screening, the ADUNU model offers a promising blueprint for other countries seeking to implement or scale up screening services [8]. As nations move forward with the mandate for universal RHD screening, further refinement of the ADUNU training model will position it as a scalable, adaptable framework that can guide efforts to reduce the burden of rheumatic heart disease worldwide.

Key messages

### What is already known on this topic

Echocardiography screening for RHD has gained support as a public health approach, but scale up of RHD screening services is complex, especially in resource-constrained settings.

### What this study adds

We performed a logic model evaluation of the Accelerating Delivery of Rheumatic Heart Disease Prevention in Northern Uganda (ADUNU) Program, a novel RHD control program, 15 months after its implementation within the Ugandan public health care system. Our proposed training model for scaling up RHD echocardiography training within an existing public health system in sub-Saharan Africa is feasible and effective, but resource intensive.

### How this study might affect research, practice, or policy

Our model is a scalable and adaptable framework that can guide efforts globally to reduce the burden of RHD.

## Supporting information

**S1 Fig. Training program outline.**
(DOCX)

**S2 Fig. Revised training program education outline.**
(DOCX)

## Author contributions

**Conceptualization:** Paul W Warren, Scott Wirth, Jafesi Pulle, Doreen Nakaagayi, Michaela Pardo, Francis Odong, Ronald Ogwanh, Isaac Otim, Craig Sable, Alison Spaziani, Emmy Okello, Kristen Danforth, Andrea Beaton.

**Data curation:** Paul W Warren, Andrea Beaton.

**Formal analysis:** Paul W Warren, Scott Wirth, Kristen Danforth, Andrea Beaton.

**Funding acquisition:** Doreen Nakaagayi, Michaela Pardo, Craig Sable, Alison Spaziani, David Watkins, Emmy Okello, Kristen Danforth, Andrea Beaton.

**Investigation:** Paul W Warren, Scott Wirth, Jafesi Pulle, Doreen Nakaagayi, Evelyn Ayaa, Michaela Pardo, Ronald Ogwanh, Craig Sable, David Watkins, Emmy Okello, Kristen Danforth, Andrea Beaton.

**Methodology:** Paul W Warren, Scott Wirth, Jafesi Pulle, Doreen Nakaagayi, Michaela Pardo, Isaac Otim, Kristen Danforth, Andrea Beaton.

**Project administration:** Jafesi Pulle, Charles Oyoo Akia, Evelyn Ayaa, Michaela Pardo, Francis Odong, Ronald Ogwanh, John Omagino, Isaac Otim, Alison Spaziani, Henry Okello Otto, Emmy Okello, Andrea Beaton.

**Resources:** Doreen Nakaagayi, Charles Oyoo Akia, Evelyn Ayaa, Michaela Pardo, Isaac Otim, David Watkins, Andrea Beaton.

**Supervision:** Paul W Warren, Charles Oyoo Akia, Evelyn Ayaa, Isaac Otim, Craig Sable, Emmy Okello, Andrea Beaton.

**Visualization:** Paul W Warren.

**Writing – original draft:** Paul W Warren, Andrea Beaton.

**Writing – review & editing:** Paul W Warren, Scott Wirth, Jafesi Pulle, Doreen Nakaagayi, Charles Oyoo Akia, Evelyn Ayaa, Michaela Pardo, Francis Odong, Ronald Ogwanh, John Omagino, Isaac Otim, Craig Sable, Alison Spaziani, Henry Okello Otto, David Watkins, Emmy Okello, Kristen Danforth.

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
