## [Decision Letter · Decision Letter 0]

9 Nov 2025

Dear Dr. Warren,

Thank you for submitting your manuscript to PLOS ONE. After careful consideration, we feel that it has merit but does not fully meet PLOS ONE’s publication criteria as it currently stands. Therefore, we invite you to submit a revised version of the manuscript that addresses the points raised during the review process.

You have to revise the article in the light of comments of both reviewers.

We look forward to receiving your revised manuscript.

Kind regards,

Rano Mal Piryani, MBBS, MCPS, DTCD, MD, Fellowship in Med Education

Academic Editor

PLOS ONE

Journal Requirements:

NIH Grant R01HL164615

5. Please amend your authorship list in your manuscript file to include author Paul Warren.

6. Please amend the manuscript submission data (via Edit Submission) to include author Paul W Warren.

7. We note that Figure 1 in your submission contain [map/satellite] images which may be copyrighted. All PLOS content is published under the Creative Commons Attribution License (CC BY 4.0), which means that the manuscript, images, and Supporting Information files will be freely available online, and any third party is permitted to access, download, copy, distribute, and use these materials in any way, even commercially, with proper attribution. For these reasons, we cannot publish previously copyrighted maps or satellite images created using proprietary data, such as Google software (Google Maps, Street View, and Earth). For more information, see our copyright guidelines: http://journals.plos.org/plosone/s/licenses-and-copyright.

Additional Editor Comments :

Dear editor

I do agree with comments of both

Authors have to do minor revision in the lights of comments of both reviewers.

Regards

Reviewers' comments:

Reviewer's Responses to Questions

**Comments to the Author**

1. Is the manuscript technically sound, and do the data support the conclusions?

Reviewer #1: Yes

Reviewer #2: Yes

2. Has the statistical analysis been performed appropriately and rigorously?

Reviewer #1: Yes

Reviewer #2: Yes

3. Have the authors made all data underlying the findings in their manuscript fully available?

Reviewer #1: Yes

Reviewer #2: Yes

4. Is the manuscript presented in an intelligible fashion and written in standard English?

Reviewer #1: Yes

Reviewer #2: Yes

Reviewer #1: This manuscript addresses a highly relevant and timely topic: the feasibility of scaling up rheumatic heart disease (RHD) screening training within the public health system in Uganda. The work is ambitious, methodologically sound, and provides valuable lessons for other low-resource, high-prevalence settings. I believe this paper deserves publication after revisions. Below, I provide detailed comments to help strengthen the manuscript.

Reviewer #2: The topic of this manuscript is highly relevant; screening for RHD is essential but is faced with low resources in many “endemic” areas. This manuscript addresses the solution to have a cost-effective scale fulfilling this gap. Generally, the manuscript is well organized, relevant, clearly written, and well supported by extensive quantitative and qualitative data. One notable strength is the use of a logic model to frame program evaluation, which helps to align implementation activities with desired results.

I have a few items that need to be clarified:

The improvement in screening sensitivity after retraining is encouraging, but you can claim that the experience of the health workers and the number of cases they handled added to this improvement. Also, I would recommend a discussion about ways of expanding the program and how it could fit into the national health system and if it may fit for use in other countries. The authors should also address the lack of long-term patient outcomes as study limitations.

Please, explain the abbreviations when used for the first time. The authors should have the text checked for small language errors and spacing.

best of luck.

**Do you want your identity to be public for this peer review?** For information about this choice, including consent withdrawal, please see our For information about this choice, including consent withdrawal, please see our Privacy Policy .

Reviewer #1: No

Reviewer #2: No

---

## [Author Response · Author response to Decision Letter 1]

7 Dec 2025

We thank the reviewers and editor for their comments which we have addressed, and which we believe improves the manuscript. Please find attached in our uploaded documents our detailed response to each comment (which is easier to review than the below responses):

Reviewer #1: This manuscript addresses a highly relevant and timely topic: the feasibility of scaling up rheumatic heart disease (RHD) screening training within the public health system in Uganda. The work is ambitious, methodologically sound, and provides valuable lessons for other low-resource, high-prevalence settings. I believe this paper deserves publication after revisions. Below, I provide detailed comments to help strengthen the manuscript.

This is an important manuscript on rheumatic heart disease (RHD) screening training in Uganda. Its an ambitious and timely work that addresses a major public health gap in Sub-Saharan Africa. The study is well-structured, with clear methodology and valuable findings. I believe this paper deserves publication, provided the following issues are carefully addressed:

Major Points for Improvement

1. Tables and Data Presentation

Several tables are overloaded with details, which may reduce readability. Please simplify by separating inputs/resources from outcomes or moving highly technical cost details to supplementary material.

We have separated the human and equipment resources into separate tables (Tables 1 and 2). The outcomes data is presented by itself in Table 3.

Ensure that percentages and denominators are always clear (e.g., specify the “n” for conditional vs final certification groups in tables).

We have more clearly specified the denominators in Table 3 and Table 4.

2. Results and Statistics

The results are descriptive, which is acceptable, but the statistical presentation can be strengthened. Confidence intervals should be added for sensitivity/specificity estimates and for major proportions (e.g., final certification rates).

We have added confidence intervals to the sensitivity/specificity estimates and positive and negative predictive values to strengthen the statistical presentation (shown in Table 4). Confidence intervals have also been added for certification rates (Table 3). Statistical analysis was performed using RStudio (methods section updated).

Consider logistic regression or stratified analysis to identify predictors of successful certification and sustained practice. This would add depth and highlight determinants of program success.

We strongly agree with the reviewer that identification of successful certification and sustained practice is important. However, we feel it is outside the scope of this current manuscript. We highlight that the identification of key determinants is undergoing in other research projects in our “Identification of early adopters/high performers” section (lines 306-310).

3. Discussion

o The discussion is somewhat repetitive in places. Please sharpen the focus by directly contrasting your findings with other large-scale RHD screening experiences (e.g., Fiji, Brazil PROVAR, etc.).

We made the discussion more concise while also highlighting that in comparison to other large scale RHD screening training programs, our study was one of the first to critically evaluate the implementation process itself, which provided us with key insights that once improved upon will result in a more effective and scalable RHD screening program.

o Expand on the implications of the low sensitivity (59%), and propose concrete strategies for improving early detection (e.g., booster training, AI-assisted screening, embedding RHD into standard nursing curricula).

We expanded upon the implications of the low sensitivity (lines 256-260). We propose frequent and repetitive practice recognizing RHD positive screening echocardiograms as the best way to improve sensitivity. We have added our revised training program as Supplementary Information Figure 2 (line 318) as it shows the increased pattern recognition practice that is required during the initial training (1b) and independent scanning (2a) periods.

o Please include regional context. There are now reports emerging from Horn of Africa, which provide valuable complementary evidence. We recommend citing:

Abdi IA, Karataş M, Öcal L, Ahmed SA, Hassan MS, Atilla K, Mohomud MF. Pattern of rheumatic heart disease among patients attending at a tertiary care hospital in Somalia: first report from Somalia. Am J Cardiovasc Dis. 2023 Oct 15;13(5):345.

Abdi IA, Karataş M, Öcal L, Elmi Abdi A, Farah Yusuf Mohamud M. Retrospective analysis of left ventricular thrombus among heart failure patients with reduced ejection fraction at a single tertiary care hospital in Somalia. Open Access Emerg Med. 2022;14:591–7.

o Including these will help frame Uganda’s experience as part of a broader East African movement toward locally generated data and scalable solutions.

We added the citation to the first paragraph of the discussion, highlighting the growing body of regional East African literature characterizing RHD and RHD training programs.

4. Conclusion

o The conclusion should be more concise and action-oriented. Currently, it repeats parts of the discussion. Instead, emphasize three key take-home messages:

1. RHD screening training is feasible in public health systems.

2. Sensitivity remains suboptimal and requires iterative training/innovation.

3. The model is scalable and should inform regional/global policies.

We have modified the conclusion to be more concise and include these three key take-home messages.

Despite these areas for improvement, I want to emphasize that this is a very strong and much-needed paper. With sharper presentation of the data, strengthened analysis, and improved discussion, this manuscript will make an important contribution to the global RHD literature. If the above revisions are approached carefully, I am confident the paper will be accepted for publication.

We thank the reviewer for their helpful suggestions that have improved the manuscript.

Reviewer #2:

The topic of this manuscript is highly relevant; screening for RHD is essential but is faced with low resources in many “endemic” areas. This manuscript addresses the solution to have a cost-effective scale fulfilling this gap. Generally, the manuscript is well organized, relevant, clearly written, and well supported by extensive quantitative and qualitative data. One notable strength is the use of a logic model to frame program evaluation, which helps to align implementation activities with desired results.

I have a few items that need to be clarified:

The improvement in screening sensitivity after retraining is encouraging, but you can claim that the experience of the health workers and the number of cases they handled added to this improvement. Also, I would recommend a discussion about ways of expanding the program and how it could fit into the national health system and if it may fit for use in other countries. The authors should also address the lack of long-term patient outcomes as study limitations.

We agree with the reviewer that increased health care worker experience was complimentary to the re-training sessions, in improving their screening sensitivity. We have added a statement acknowledging the important role of increased healthcare worker experience and repetitions to increase sensitivity in our “Proposed modifications for the next program iteration” section (lines 329-331).

We are currently expanding the program to a second district in northern Uganda. For the model to fit into the national health system in Uganda and other countries, we must first improve the sensitivity and reduce the burden on human resources. We have specified this in our “Proposed modifications for the next program iteration” section (lines 339-343).

We have added a statement acknowledging the limitation of the current analysis due to lack of long-term data (line 327).

Please, explain the abbreviations when used for the first time. The authors should have the text checked for small language errors and spacing.

best of luck.

We have ensured that all abbreviations are explained with the first use and that language/spacing errors are eliminated.

Clarifications for the Editor

The correct grant identification is: R01HL1646150. This is what is listed in the manuscript file. We have added the requested statement regarding role of the funder to the funding statement on the title page.

Paul W Warren is listed as an author in the manuscript file and author submission form.

Captions have been uploaded for the supporting information files at the end of the manuscript.

---

## [Decision Letter · Decision Letter 1]

15 Feb 2026

Evaluation of a Training Program for Rheumatic Heart Disease Screening Integrated into the Public Health System in Uganda

PONE-D-25-39629R1

Dear Dr. Warren,

We’re pleased to inform you that your manuscript has been judged scientifically suitable for publication and will be formally accepted for publication once it meets all outstanding technical requirements.

Kind regards,

Rano Mal Piryani, MBBS, MCPS, DTCD, MD, Fellowship in Med Education

Academic Editor

PLOS One

Additional Editor Comments (optional):

Dear Authors

Greetings

I appreciate that you have well responded to reviewer' comments point-wise.

With best regards

Reviewers' comments:

Reviewer's Responses to Questions

**Comments to the Author**

Reviewer #1: All comments have been addressed

Reviewer #2: All comments have been addressed

2. Is the manuscript technically sound, and do the data support the conclusions?

Reviewer #1: Yes

Reviewer #2: Yes

3. Has the statistical analysis been performed appropriately and rigorously?

Reviewer #1: Yes

Reviewer #2: Yes

4. Have the authors made all data underlying the findings in their manuscript fully available?

Reviewer #1: Yes

Reviewer #2: Yes

5. Is the manuscript presented in an intelligible fashion and written in standard English?

Reviewer #1: Yes

Reviewer #2: Yes

Reviewer #1: The authors have addressed all previous comments, and the manuscript has been clearly improved. I have no further comments and consider it suitable for publication.

Reviewer #2: The authors have adequately addressed the comments, and I recommend acceptance of the manuscript in its revised form. Good luck

**Do you want your identity to be public for this peer review?** For information about this choice, including consent withdrawal, please see our For information about this choice, including consent withdrawal, please see our Privacy Policy .

Reviewer #1: No

Reviewer #2: No

---

## [Editor Report · Acceptance letter]

PONE-D-25-39629R1

PLOS One

Dear Dr. Warren,

I'm pleased to inform you that your manuscript has been deemed suitable for publication in PLOS One. Congratulations! Your manuscript is now being handed over to our production team.

Kind regards,

on behalf of

Dr. Rano Mal Piryani

Academic Editor

PLOS One